# Maximizing the Production of Recombinant Proteins in Plants: From Transcription to Protein Stability

**DOI:** 10.3390/ijms232113516

**Published:** 2022-11-04

**Authors:** Ziru Feng, Xifeng Li, Baofang Fan, Cheng Zhu, Zhixiang Chen

**Affiliations:** 1College of Life Sciences, China Jiliang University, Hangzhou 310018, China; 2Department of Botany and Plant Pathology, Purdue Center for Plant Biology, Purdue University, West Lafayette, IN 47907-2054, USA

**Keywords:** recombinant protein production, plant expression system, transient expression, viral expression vectors, RNA silencing, subcellular protein targeting, inducible promoters

## Abstract

The production of therapeutic and industrial recombinant proteins in plants has advantages over established bacterial and mammalian systems in terms of cost, scalability, growth conditions, and product safety. In order to compete with these conventional expression systems, however, plant expression platforms must have additional economic advantages by demonstrating a high protein production yield with consistent quality. Over the past decades, important progress has been made in developing strategies to increase the yield of recombinant proteins in plants by enhancing their expression and reducing their degradation. Unlike bacterial and animal systems, plant expression systems can utilize not only cell cultures but also whole plants for the production of recombinant proteins. The development of viral vectors and chloroplast transformation has opened new strategies to drastically increase the yield of recombinant proteins from plants. The identification of promoters for strong, constitutive, and inducible promoters or the tissue-specific expression of transgenes allows for the production of recombinant proteins at high levels and for special purposes. Advances in the understanding of RNAi have led to effective strategies for reducing gene silencing and increasing recombinant protein production. An increased understanding of protein translation, quality control, trafficking, and degradation has also helped with the development of approaches to enhance the synthesis and stability of recombinant proteins in plants. In this review, we discuss the progress in understanding the processes that control the synthesis and degradation of gene transcripts and proteins, which underlie a variety of developed strategies aimed at maximizing recombinant protein production in plants.

## 1. Introduction

Proteins, which are made of one or more chains of amino acid residues, are tremendously diverse in both their structures and functions. Proteins perform a vast spectrum of structural, biochemical and molecular activities within organisms. Many of these biological functions of proteins also have important medical, industrial and scientific applications. By 2018, 374 protein-based pharmaceutical products had gained a license in the United States and European Union, which accounts for about a third of all pharmaceuticals in development [1]. These protein-based pharmaceutical products include monoclonal antibodies, hormones, enzymes, vaccines, clotting and growth factors. Recombinant proteins including enzymes have also been extensively used in production of textiles and chemicals and in processing of food and feed. Other recombinant proteins have been widely used in diagnostics and scientific research. The global protein expression market size was US$1.65 billion in 2017 and was projected to increase at more than 10% annually to reach US$6.47 billion by 2030 [2,3].

Most recombinant proteins are currently produced in bacteria (mainly *Escherichia coli*) and mammalian cells (e.g., Chinese hamster ovary cells). Bacterial cells are easy and cheap to culture but lack the capacity to perform certain post-translational modifications. As a result, many complex proteins particularly those therapeutic antibodies and vaccines are usually produced in mammalian cells [2,3]. With the development of plant transformation in early 1980s [4,5], the potential to produce recombinant proteins in plants has also been extensively explored. Plants and plant cells have a number of advantages over bacterial and mammalian platforms for production of recombinant proteins including therapeutic proteins [6]. These advantages include the low cost and relatively high speed of recombinant protein production at a large scale in plants and plant cells [7,8]. As eukaryotic organisms, plants can carry out many of the post-translational modifications for production of complex proteins but do not require animal derived reagents (serum-free) for cultivation [9]. Plants also lack harmful or toxic substances present in bacteria and many therapeutic proteins and injectable and edible vaccines do not require extensive purification when produced in plants [9,10,11]. Despite these advantages, the number of recombinant proteins produced in plants for medical and industrial applications is still very small when compared to those produced in bacterial and mammalian systems. A recent survey has listed fewer than 40 plant-derived recombinant products and their clinical status [12]. Among the problems that limit the competitiveness of plants and plant cells for recombinant protein production are the low yield due to relatively low levels of synthesis and high levels of degradation when compared to bacterial and mammalian cells [3,13]. In addition, there are some differences in protein glycosylation in term of N-glycan composition between plants and animals that can affect the distribution, stability, activity and immunogenicity of therapeutic proteins [13].

Like biosynthesis of native proteins, production of recombinant proteins in a heterologous expression system is a complex process of gene expression. The levels of proteins as the products of gene expression are influenced by a multitude of processes including transcription, mRNA stability, translation, protein modification and folding, protein trafficking and degradation. Over the past several decades, extensive research has provided extensive knowledge about the regulation of gene expression at both transcriptional and post-transcriptional levels that is fundamental to the understanding of plant growth, development and responses to environmental conditions. Such knowledge from the fundamental research has been exploited to develop strategies to increase production of recombinant proteins in plants and plant cells. For example, identification of strong constitutive promoters enables production of recombinant proteins at high levels in plants, while inducible promoters allow production of potentially toxic proteins and tissue-specific promoters can lead to accumulation of recombinant proteins in certain plant tissues such as seeds for special purposes [3,14]. The extraordinary progress in the field of RNA interference (RNAi) makes it possible to design strategies to reduce gene silencing to increase production of recombinant proteins [3,15,16]. Major advances in the understanding of the molecular mechanisms of protein translation, quality control, trafficking and degradation have also helped development of strategies that enhance synthesis and stability of recombinant proteins in plant cells [17]. In this review, we first briefly discuss the major types of plant systems and the DNA vectors that have been developed for recombinant protein production in plants. The major focus of the review, however, will be on the progresses in the understanding of the processes and mechanisms that control the synthesis and degradation of gene transcripts and products (proteins) that underlie a variety of strategies developed to increase accumulation of recombinant proteins in plants and plant cells. 

## 2. Major Types of Plant Systems and Vectors for Recombinant Protein Production

Cell culture, stable transgenic plants and transiently expression are three major systems for production of recombinant proteins in plants (Figure 1). Plant cell culture as an expression system for recombinant proteins is similar to those of bacteria (e.g., *E. coli*) and mammals (e.g., Chinese hamster ovary cells). Although cell cultures from plants including carrots, tomato, Arabidopsis, tobacco and rice have been used for expression of heterologous proteins, tobacco BY2 (bright yellow 2) cell culture is the most commonly used system for recombinant protein expression because it grows vigorously with less tendency to aggregate and can be easily transformed genetically with various expression vectors including those from plant viruses [18,19,20]. Plant cell expression system has been used for production of Elelyso (taliglucerase alfa), the first plant-cell-expressed enzyme approved by the US Food and Drug Administration for long-term enzyme replacement therapy for type 1 Guaucher disease [21]. When compared to the microbial and mammalian systems, plant cell culture has several advantages as a platform to produce recombinant proteins [22,23,24]. Bacterial cells do not have the ability to perform complex posttranslational modifications including glycosylation and disulfide bond formation and correct folding required for the activity of many recombinant proteins. Some bacteria also produce endotoxins that could lead to healthy risk if not removed from the biopharmaceutical proteins. Mammalian cells, on the other hand, require complicated and expensive culture conditions with growth media containing animal-derived components and have the rick of harboring human pathogens. Plant cells are inexpensive to grow, capable of carrying out many of the posttranslational modifications as human cells but free of human pathogens and bacterial endotoxins. Plant cell culture also has the advantage over whole plants for purification of recombinant proteins if they are secreted into the growth media. On the other hand, unlike whole plants, plant cell culture has the disadvantages of increased equipment costs and the difficulty to be changed in capacity or scale in the recombinant protein production.

Whole plants can be used to produce recombinant proteins either through stable transformation or transient expression of genes encoding recombinant proteins to be produced (Figure 1). Stable transgenic plants with expression vectors inserted into the nuclear or chloroplast genomes have been used to produce recombinant proteins (Figure 1). Stable nuclear expression of transgenes in plants is the most widely used method for genetically engineering plants and can be potentially used for production of recombinant proteins in plants at a large agriculture scale in the field. By using tissue-specific promoters as discussed later, it is also possible to target accumulation of recombinant proteins in specific tissues or organs such as seeds and fruits for special purposes (e.g., production of edible vaccines and other recombinant proteins in seeds and fruits) [25,26,27]. Chloroplast expression of transgenes can produce higher yield of recombinant proteins than nuclear expression because of the large number of chloroplasts in each cell and lack of gene silencing [28,29,30]. Chloroplast expression of transgenes also has reduced risk of gene transfer to the environment because of its rare transmission through pollens. On the other hand, like bacteria, chloroplasts cannot perform glycosylation. Generating transgenic plants that produce stable and high levels of recombinant proteins is also very time-consuming with both lengthy technical and regulatory processes. Although it has been relatively routinely performed in tobacco and several other Solanaceous species, successful chloroplast transformation in other plant species is more limited.

Unlike stable transgenic plants, the transient expression system is rapid for production of recombinant proteins in plants (Figure 1) [31,32,33]. Transient expression of foreign genes has been reported in a variety of plants but is most widely carried out in *Nicotiana benthamiana* because of easy manipulation and the high levels of protein production. The rapidity of the system makes it particularly useful for rapid production of specific recombinant proteins in response to an emergency such as vaccines during a pandemic. Transient gene expression for recombinant protein production can be mediated by infiltration with *Agrobacterium tumefaciens* (agroinfiltration) harboring a binary vector containing the gene construct as part of the T-DNA to be transferred to the plant cells for transient expression [34] (Figure 1). Transient expression using agroinfiltration often yields much higher levels of recombinant proteins that the stably integrated genes in transgenic plants [35,36]. Transgenes to be expressed can also be engineered into the genomes of plant viruses such as *Tobacco mosaic virus* (*TMV*) and introduced into infected plants [37] (Figure 1). This procedure, however, has two major limitations. First, the system requires infectious recombinant viruses or a full copy of viral DNA/RNA to deliver the transgenes into the plant cells for expression. These viral vectors often have a size restriction on a transgene and are often inefficient in infecting plant cells. Second, production of mature viral particles including virus coat proteins in infected plants not only utilizes plant host resource but also poses potential biohazard for production of recombinant proteins. To overcome these problems, new deconstructed viral vectors have been developed [38]. These new viral vectors are Agrobacterium binary vectors whose T-DNA regions contain viral replicons with the transgenes to be expression but devoid of those limiting or undesirable viral genes [31,38,39,40,41,42,43] (Figure 1). Thus, viral replicons are transferred to the plant nucleus by transfer of T-DNA after agroinfiltration to provide rapid, systemic and efficient infection. The viral machinery, on the other hand, once delivered into plant cells drive the massive viral RNA replication and production of recombinant proteins. This combination of agrobacterial binary vectors and recombinant viral replicons allows for massive production of recombinant proteins with the need to generate transgenic plants.

## 3. Elements of Genetic Constructs for Transcriptional Enhancement and Control

A critical component of a genetic construct for production of recombinant proteins in plants or plant cells is its promoter that controls gene transcription. The promoter in the expression construct largely determines the spatial and temporal patterns of the levels of mRNA of the target gene. For increased transcription of a target gene in a non-inducible, non-tissue specific manner, strongly constitutive promoters such as the *CaMV 35S* promoter are often used for the production of recombinant proteins [44,45] (Figure 2). However, the *CaMV 35S* promoter has been mostly used for high levels of expression of target genes in the shoots of dicotyledonous plants [46,47,48]. There are other promoters such as ubiquitin promoters that have been extensively used for high-level expression of transgenes in monocotyledonous plants [49]. In addition to promoters, other *cis*-acting elements collectively known as enhances can increase the level of transcription. An enhancer may be present upstream or downstream of a gene. The function of an enhancer is often not affected by orientation. Enhancers may also be located within introns. Introduction of a plant or synthetic enhancer or enhancer intron can further increase the transcription of a target gene [50,51,52,53]. 

Plant seeds offer a number of advantages for production of recombinant proteins [13,25,27]. First, seeds naturally can accumulate a massive amount of storage proteins. Second, the dormancy properties of seeds provide stability of produced recombinant proteins for long-term storage, thereby decoupling the processing from the growth and harvest processes. Third, the low content and water and other biomolecules also reduce cost of the manufacturing processes for the production of recombinant proteins. Seed-specific and seed-restrictive promoters are commonly used to derive gene expression for recombinant proteins in seeds [54] (Figure 2). In dicotyledonous plants, promoters from storage protein genes in legumes have been widely used in transgenic plants. For example, the promoter from the phaseolin gene drives strong and seed specific expression of transgenes in dicotyledonous plants including Arabidopsis [54,55]. In monocots (primarily rice and maize), the constitutive ubiquitin promoter can drive expression in both endosperm and scutellum at relatively low levels [56]. On the other hand, globulin-1 gene promoters from rice and maize have been successfully used for high levels of seed-specific expression of foreign genes for production of recombinant proteins in seeds [57,58]. To promote production of recombinant proteins in both endosperm and embryo, it is also desirable to have two separate expression cassettes for the target gene controlled by two different promoters, one endosperm-specific and the other embryo-specific [54,59]. 

Some recombinant proteins, particularly when produced at high levels, are toxic or harmful to plant growth and development and, therefore, inducible expression of their genes is often desirable or even required for production of their products. Promoters of many plant genes confer inducible expression of transgenes in response to a variety of stimuli including pathogens, hormones and stress conditions. However, regulated expression of target genes conferred by these promoters is usually neither very tight with high background nor highly inducible. Over the past two decades or so, a number of chemical-inducible promoters have been developed from heterologous elements from animal and microorganisms that direct highly inducible expression of transgenes in plants [60] (Figure 2). The ethanol inducible gene expression system, for example, is based on the AlcR transcription factor and its target promoter *AlcA* in the fungus *Aspergillus nidulans* with a great potential for large-scale application because ethanol is cheap with relatively low toxicity and can be easily applied to plants [61,62,63,64] (Figure 3).

As discussed earlier, new unconstructed virus vectors have been successfully used for production of recombinant proteins through massive transient expression of transgenes as part of viral RNA replication. The system has been further modified for inducible release of virus replicons so high levels of recombinant protein production can be achieved in stable transgenic plants (Figure 3). For example, Werner and colleagues developed a deconstructed and double-inducible *TMV* vector in which both the replicon and movement protein are placed separately and under the control of the ethanol-inducible promoter [65]. As a result, both the replicon activation and spread of the virus in the transgenic plants are under tight control before application of ethanol to ensure normal plant growth and development. Using a similar strategy, Dugdale and colleagues developed the In Plant Activation (INPACT) system based on the replication machinery of *Tobacco yellow dwarf mastrevirus* (*TYDV*), a dicot-infecting single DNA virus [66,67]. The INPACT system contains two cassettes: one containing the viral Replication/Replication A (Rep/RepA) genes under the transcriptional control of the *AlcA:AlcR* gene switch and the INPACT cassette containing the gene of interest and necessary cis-acting sequences for replication recognition (Figure 3). In transgenic plants containing the two INPACT cassettes, application of a low level of ethanol leads to induction of Rep/Rep A genes and production of the virus-encoded replication activation proteins. TYDV Rep and RepA recognize the replication cis-acting elements of the integrated INPACT cassette and use it as a template for rolling circle replication to produce single-stranded circular DNA copies (episomes) of the INPACT cassette (Figure 3). After converting to double stranded DNA molecules by host polymerases, the double-stranded episomes can serve as templated for further replication and become transcriptionally active for production of transgene mRNAs. Therefore, unlike the TMV system, the INPACT system does not produce viral RNAs capable of replication and cell-to- cell movement. Furthermore, TYDV has a broader host range than TMV and therefore the INPACT technology can be applied to a broad range of plant species [66,67].

There are additional strategies for promoting transcription of transgenes in plants. Damaj and colleagues have recently reported a combinatorial promoter stacking system to increase production of recombinant proteins in sugarcane culms [68]. In the reported study, the gene encoding the antimicrobial Bovine lysozyme was under control of multiple constitutive and culm-specific promoters in separate expression vectors and transformed into sugarcane using particle bombardment. The levels of the recombinant proteins accumulated at high levels in lines containing the stacked promoter: *BvLz* constructs, which could be further enhanced by re-transformation with additional expression vectors [68]. Accumulation of high levels of recombinant proteins in these lines was stable in multiple vegetative propagations. In rice, the sugar starvation-inducible *αAmy3* promoter and its signal peptide has been widely used for production of recombinant proteins in rice suspension cells [69]. Rice MYBS1 is a transcription factor that recognizes specific cis-acting elements in the *αAmy3* promoter to activate the inducible expression of the endogenous *αAmy3* gene. Rice MYBS2, on the other hand, is a negative regulator of *αAmy3* expression by competing for binding sites with MYBS1 in the *αAmy3* promoter [69]. By knocking down the expression of MYBS2 expression in rice suspension culture, Sinaga and colleagues were able to increase accumulation of recombinant proteins by 2-5-fold [69].

Genetic transformation, through direct or Agrobacterium-mediated methods, randomly inserts transgenes into the nuclear genome of plant cells with single or multiple copies at the same or different loci. Due to the difference in insertion sites and copy number, the expression level of transgenes is highly variable due to positional effects and gene silencing. Inclusion of specific DNA sequences such as matrix attachment regions (MARs) and genetic insulators have been shown to reduce positional effects, suppress chromatin silencing and increase transgene expression [70] (Figure 2). MARs are AT-rich DNA sequences that are believed to bind nuclear matrix and promote organization of chromosomal DNA in transcriptionally active states [71,72,73]. Genetic insulators are DNA cis-acting elements that can inactivate the effect of distal or nearby enhancers, blockers and promoters on the expression of a gene when placed between the genes and the regulatory sequences of transcription [70]. NI29 is a genetic insulator from Arabidopsis with a 16-bp palindromic sequence. The M14 element is a modified derivative of NI20 with enhanced insulating activity [70,74]. 

## 4. Minimizing Post-Transcriptional Gene Silencing (PTGS)

PTGS is an RNA silencing mechanism that degrades specific mRNAs and reduces the expression of a specific gene [75,76,77]. RNA silencing is triggered by dsRNAs and therefore can be induced efficiently by expressing transgenes with inverted repeats or in the antisense orientation [15]. RNA silencing can also be induced frequently by sense transgenes designed for overexpression (Figure 2). In fact, as a genome surveillance system, RNA silencing detects and eliminates transcripts from excessively expressed genes including transgenes under control of strong promoters [78,79,80,81]. RNA silencing of sense transgenes in Arabidopsis requires RDR6 (RNA-dependent RNA polymerase 6) (also known as SDE1 or SGS2), which may recognize certain aberrant RNAs of silenced transgenes as templates for synthesis of dsRNA to trigger RNA silencing [82] (Figure 2). Using GUS reporter gene or gene repeats with or without 3′ transcription terminators, we have provided strong evidence that improperly terminated, unpolyadenylated mRNA from transgene transcription is the long-sought aberrant RNAs that are recognized by and act as templates of RDR6 for dsRNA synthesis to trigger RNAi [83] (Figure 2). This hypothesis has since been supported by direct assays of purified recombinant RDR6 using RNAs with or without poly(A) tails as substrates [84]. Apparently, when a transgene is highly expressed, there will also be increased levels of unpolyadenylated mRNAs due to abortive transcription elongation, the premature termination of transcription, or mRNA 3′ readthrough.

Commonly used transcription terminators are leaky when used with a transgene driven by a strong promoter, leading to mRNA 3′ readthrough and production of unpolyadenylated mRNAs, which trigger RNAi [83,85]. Based on these findings, we have previously tested possible enhancement of transgene expression by using double terminators (Figure 2). When the GUS reporter gene is driven by the strong *CaMV 35S* promoter, its expression can be enhanced by 3-4-fold if its 3′ end contains both the *CaMV 35S* and Agrobacterium *nos* terminators, when compared with the *35S* or *nos* single terminator [83]. A substantial number of groups have since found similar enhancement of transgene expression by using double terminators in the expression cassettes [86,87,88,89,90,91,92] (Figure 2). Beyene and colleagues, for example, reported that in sugarcane biolistic transformation, expression of a yellow fluorescent protein (eYFP) reporter gene driven by the maize ubiquitin promoter can be enhanced by up to more than 100-fold by a double terminator when compared to single terminators [86]. In addition, there has been effort to identify better terminators for enhancing transgene expression and production of recombinant proteins. Diamos and colleagues screened 20 commonly used or newly identified terminators and discovered 8 to be better than the *35S* and *nos* terminators [93]. The extensin terminator provided more than 13-fold increase in transgene expression than the *nos* terminator, while combining terminator in tandem provided 25-fold increase [93]. Other elements such as MARs were added to the combined terminators to provide further enhancement of transgenes [93]. When placed in a replicating geminiviral vector, the system was used for high level production of monoclonal antibodies [94].

PTGS functions as an antiviral mechanism by eliminating viral transcripts or genomes [16]. As a countermeasure to the defense mechanisms, viruses have evolved to encode proteins that suppress PTGS (Figure 2). Among the best characterized suppressors of PTGS are the helper component-proteinase (HC-Pro) protein of potyviruses and P19 from *Tomato bushy stunt virus* (*TBSV*) [16]. These viral PTGS suppressors have been used to increase production of recombinant proteins in plant cells (Figure 2). A number of groups reported increased expression of recombinant proteins through transient co-expression of viral PTGS suppressors in *N. bentamiana*. Arzola and colleagues tested 9 viral suppressors including HC-Pro and P19 and found that co-expression of some of the viral proteins in *N. benthamiana* could increase the expression of a novel anthrax receptor decoy protein by 10-fold under control of the *CaMV 35S* promoter [95]. The same research group has more recently reported enhancement of recombinant protein production in *N. benthamiana* cell suspension cultures by co-cultivation of Agrobacterium containing viral silencing suppressors [96]. Nine viral PGTS suppressors were tested individually or in combination were tested in transgenic plant cell culture for production of recombinant human alpha-1-antitrypsin under control of the constitutive *CaMV 35S* promoter or an estrogen-inducible promoter. The results showed that in transgenic cell cultures, production of the recombinant protein increased by almost 6-fold with the expression of P19 viral suppressor and more than 17-fold with the co-expression of P19 and two other suppressors [96]. Garabagi and colleagues reported use of P19 viral PTGS suppressor in *N. benthamiana* to increase production of Trastuzumab, a therapeutic antibody used in the treatment of HER2+ breast cancer [97]. Habibi and colleagues reported increased production of the HIV-1 entry inhibitor griffithsin through transient expression in *N. bethamiana* by co-expressing a combination of three viral PTGS suppressors, P19 from *TBSV*, P1 from *Rice yellow mottlevirus* (*RYMV*) and P0 from *Beet western yellow virus* (*BMYMV*), which inhibit different targets in the RNAi silencing pathways [98].

## 5. Enhancing the Translation of Recombinant Proteins

The efficiency of mRNA translation is a critical factor that determines the levels of recombinant protein production in plants. In eukaryotes, most mRNAs are translated by a scanning mechanism by which the 43S translation preinitiation complex is first attached to the free 5′ end of the mRNA and scan base by base for the translation initiation codon [99,100]. Thus, the first AUG encountered is favored as the primary initiation site. Thus, the 5′-UTR of transgene mRNAs should avoid potential secondary structures, which could interfere with the movement and scanning by the translation preinitiation complex and reduce translation efficiency [101]. In addition, alternative upstream AUGs can be recognized as primary but false initiation codons and reduce the translation efficiency of the protein-coding part of mRNAs, and, therefore, should also be avoid when designing the expression cassettes for recombinant protein production. On the other hand, specific sequences immediately surrounding the translation AUG initiation codon, particularly at the −3 and +4 positions, affect the efficiency of translation of transgene mRNA [102]. 

The translation of plant RNA virus genomes is highly efficient. Many plant viral RNAs do not contain a 5′ cap and rely on internal ribosome entry site (IRES) in the 5′-UTR or a cap-independent translation element (CITE) in the 3′-UTR to promote translation [103]. Many plant RNA viruses also lack the poly(A) tail at the 3′ end but have evolved specific structures to replace the function of a poly(A) tail, which is also required for translation initiation [103]. Therefore, plant RNA viruses contain a variety of translation enhancer sequences that stimulate viral RNA translation to the highest levels. Some of these viral translation enhancer elements have been incorporated into transgenes for expressing high levels of recombinant proteins in plant cells. The omega (Ω) sequence from Tobacco mosaic virus (TMV) is one of the most efficient translation enhancers for enhancing translation when inserted into the 5′-UTR of transgenic mRNAs [103,104] (Figure 2). The Ω sequence is located within the 68bp 5′-leader sequence of the TMV genomic mRNA, and can enhance translation by facilitating ribosome recruitment through recognition by the heat shock protein, HSP101. HSP101-mediated translation enhancement of omega sequence-containing mRNA require two eukaryotic translation initiation factors (eIFs), eIF4G and eIF3, which ae known to promote the recruitment of 40S ribosomal subunits to an mRNA [105,106].

Codon optimization of the coding sequences of transgenes is another commonly used approach to increase production of recombinant proteins [13,107,108] (Figure 2). Due to the degenerate nature of the genetic code, most amino acids are encoded by multiple synonymous codes. However, cells from an organism do not express all tRNAs for all possible genetic codons and those expressed tRNAs vary drastically in the relatively expression levels. Different organisms also differ highly in the relative levels of individual tRNAs. Accordingly, it is generally assumed that rare codons may have relatively low levels of corresponding tRNAs and are rating-limiting for protein production and therefore, replacing them with frequently used codons increases protein production [108,109]. However, the relationship between the frequencies and synonymous codons and protein production is complex and has not been fully established in eukaryotes. In some reported studies, altering the frequencies of codon usages have been shown to affect recombinant protein production in plants. For example, when the herbicide-resistant *bar* genes from *Stretomyces hygroscopicus* with different percentage and placement of optimal codons were analyzed in transgenic tobacco lines, it was found that a certain percentage (up to 54%) of optimal codons increased transgene expression [110]. However, further increase in the percentage of optimal codons did not lead to further improvement in protein expression [110]. A similar enhancement of protein expression has been observed for recombinant proteins expressed through chloroplast genomes in transplastomic tobacco and lettuce lines [111]. Codon optimization of human clotting factor and polio viral capsid protein genes based on plant *psbA* gene codon usage increased production of these recombinant proteins by 5-30-fold [111]. Further analysis indicated that the increase of codon optimized protein synthesis is at the translational level and associated with reduced ribosome pause upon codon optimization [111].

## 6. Subcellular Localization of Recombinant Proteins to Promote Their Accumulation

Once recombinant proteins are synthesized in plant cells, their subcellular localization is a key factor that influences their folding, quality control, modification, and degradation, which determine their stability and quality [112]. In the absence of any subcellular targeting signal, a nuclear-encoded recombinant protein remain in the cytosol and usually accumulates at low levels even when its mRNA levels are high. Low accumulation of some recombinant proteins in cytosol may be related to their special need for posttranslational modification such as disulfide bridge formation due to unsuitable redox potential in the subcellular compartment. The presence of the ubiquitin-dependent proteasome system in the cytosol could also reduce the stability of recombinant proteins. Targeting a recombinant protein to some subcellular compartments such as the apoplast, chloroplasts, vacuoles, specialized endoplasmic reticulum (ER)-derived vesicles can drastically increase the yield of the recombinant protein (Figure 4). For example, targeting a human growth hormone to the apoplast can increase it accumulation by about 1000-fold when compared to its accumulation in the cytosol [113]. Targeting recombinant proteins to the extracellular space in plant cell suspension culture also facilitate their purification.

Plants accumulate high levels of storage proteins in the vacuole, which has also been exploited as a subcellular destination for the deposition of recombinant proteins (Figure 4). In plant storage organs such as seeds, storage proteins are synthesized on the rough ER and transported to the protein storage vacuoles (PSVs) through several pathways [114]. In the receptor-mediated sorting pathway, vacuolar sorting of storage proteins is mediated by specific protein-protein interactions between transmembrane vacuolar sorting receptors (VSRs) and the vacuolar sorting determinants (VSDs) located on the storage cargo proteins [115]. By using seed-specific promoters and VSDs, a substantial number of recombinant proteins including growth hormones, lysozymes, insulin and antibodies have been stably produced in the seeds of different plants [25] (Figure 4). Vegetative tissue vacuoles are hydrolytic and are generally not considered to be suitable for deposition of recombinant proteins. However, when fused with specific VSDs, some recombinant proteins including avidin, cellulolytic enzymes, endolysin, transglutaminases, several secretory mammalian proteins such as collagen, α1-proteinase inhibitor, interleukin-6 and immunoglobulins accumulate at high levels in leaf vacuoles [116]. Enhanced accumulation of recombinant proteins fused to VSDs in vegetative tissue vacuoles has been demonstrated in several plants including tobacco, *N. benthamiana*, sugarcane, tomato and carrot [116]. Therefore, the hydrolytic nature of vacuoles in vegetative tissues appears to be highly dynamic and is strongly influenced by ectopic deposition of proteins. 

Upon synthesis on the ER, some storage proteins such as zeins in maize accumulate in specialized ER-derived protein bodies, which are widely present in cereal plants and have also been studied as storage organelles for recombinant protein production in plant cells [117] (Figure 4). Protein bodies can be induced artificially in plant leaves by overexpression of recombinant proteins fused with a protein body-inducing tag such as Zera, elastin-like polypeptide (ELP) and hydrophobins (HFBs) [32,118]. Zera is a peptide of 112 residues composed of the signal peptide and N-terminal proline-rich domain of γ-zein with six cysteine residues capable for forming inter-chain disulfide bonds [119,120,121,122]. Zera fusions have been used for production of recombinant human growth hormone, epidermal growth factor and Streptomyces derived xylanases [121,123]. ELPs are synthetic biopolymers with VPGXG repeats (X can be any non-proline amino acid) originally identified in the mammalian protein elastin [124,125]. ELPs are intrinsically disordered proteins (IDPs) and undergo a reversible phase transition from soluble protein to insoluble aggregates above specific transition temperatures, which facilitates rapid protein purification through inverse transition cycling [126]. ELP peptides increases accumulation of recombinant spider silk proteins, murine interleukin-4 [127], human interleukin-10 [128], anti-HIV antibody 2F5 [129], and neutralizing antibodies against H5N1 virus [130]. In *N. benthamiana* leaves, ELP increased GFP accumulation (up to 40% of total soluble proteins) by inducing formation of GFP-containing protein bodies [131]. HFBs are a family of small, secretory proteins produced by filamentous fungi with extraordinarily surface-active properties [132], which can be transferred to recombinant proteins and used for purification using aqueous two-phase separation [133]. HFB1 from *Trichoderma reesei* can increase the accumulation of glucose oxidase, which is difficult to express with other expression systems [134]. HFBI as a fusion tag also improved accumulation of GFP up to 51% of the total soluble protein and increased the yield of other target proteins in plants [134].

Chloroplasts are one of the most attractive organelles for production of recombinant proteins because of the number and the area in mesophyll cells, neutral pH and low levels of proteolysis [135] (Figure 4). Two different approaches have been used for production of recombinant proteins in chloroplasts. The first approach is based on introduction of transgenes into the chloroplast genome through homologous recombination following gene delivery commonly by the biolistic method. This approach has a number of advantages including high-level expression due to the high number of chloroplasts and chloroplast genomes per chloroplast, site-specific gene integration and reduced transgene spread through pollens because of maternal inheritance of plastids. In addition, because of the polycistronic chloroplast gene expression, it is possible to express multiple recombinant proteins from a polycistronic mRNA in transplastomic lines. The second approach is based on nuclear transformation of transgenes, which can either expressed transiently in plant cell nucleus or stably inserted into the plant nuclear genome (Figure 4). The translated proteins with an N-terminal terminal transit peptide and imported into chloroplasts through translocon complexes in the outer (TOC) and inner (TIC) envelope membrane. Because these chloroplast-destinated recombinant proteins are synthesized in the cytosol, their accumulation in chloroplasts is subjected to influence by chloroplast protein import capacity and ubiquitin-dependent turnover of their precursors in the cytosol [136]. In addition, unlike ER-derived structures, chloroplasts lack the ability for glycosylation of proteins, which is important for both the stability and activity of some recombinant proteins. However, some chloroplast proteins including a-type carbonic anhydrases and amylases are N-glycosylated and accumulate at high levels in chloroplasts [137]. These N-glycosylated chloroplast proteins are first targeted to the ER for N-glycosylation, further modified in the Golgi apparatus and then imported into chloroplasts [106,138,139]. Elucidation of the mechanism by which N-glycosylated proteins are trafficked into chloroplasts can lead to development of strategies to production of N-glycosylated recombinant proteins in chloroplasts. 

## 7. Summary and Prospects

The production of recombinant proteins in plants and plant cells has important advantages as well as challenges. A major challenge for making plants into a competitive platform for recombinant protein production is understanding how to elevate protein yield by enhancing expression and reducing degradation. Over the past several decades, important developments in our understanding of the mechanisms by which gene expression is regulated in plants have led to the development of a wide range of tools, methods, and strategies for achieving high and consistent yields of recombinant proteins from plants. A variety of regulatory DNA elements, including promoters, terminators, enhancers, and genetic insulators, are now available for the strong and stable expression of transgenes in plants and plant cells. New methods have been developed for the massive production of transgene transcripts and recombinant proteins through the inducible release of viral replisomes. Effective strategies have also been reported for increasing the stability of recombinant proteins by targeting them to specific subcellular compartments, particularly the storage organelles, including PSVs and protein bodies. Using these developed methods and strategies, several groups have reported the successful production of recombinant proteins with exceptionally high yields of up to more than 50% of the total leaf proteins.

Despite these critical developments, there are still important challenges for large-scale plant-based production of recombinant proteins. First, successful production of recombinant proteins at very high levels using some of the developed strategies has been demonstrated for only a very limited number of recombinant proteins and it is unclear whether they are as effective for a wide range of recombinant proteins that differ in structures, size, folding and stability. It is likely that many of these developed tools, methods and strategies require further improvement and optimization for successful production of those recombinant proteins that are difficult to produce in plants. It may also be necessary to integrate many of these developed tools and methods to achieve strong and stable production of recombinant proteins in plants. Second, many of the successful studies on the high-level production of recombinant proteins have been limited to plant leave tissues (primarily in *N. benthamiana*). However, some other tissues and organs such as seeds may offer special advantages including better potential for molecular farming for production of recombinant proteins. Even though a great deal has been known about the synthesis and trafficking of seed storage proteins, there are still important challenges for production of stable and functional recombinant proteins at high abundance in seeds. Third, production of recombinant proteins in plants also face the issues of inconsistent protein quality and difficulties for large-scale downstream processing, both of which are closely associated with the efforts to increase protein yield. For example, targeting a recombinant protein to a particular subcellular compartment may affect its posttranscriptional modifications, which may determine its activity and quality. Fusion of a recombinant protein with a protein body-inducing tag could drastically increase protein accumulation but may affect downstream processing if the tag affects the activity and quality of the recombinant proteins. Finally, an important challenge for production of recombinant proteins in plants is the potential spread of recombinant genes as with all other genetically modified plants [13]. Some of the strategies for increasing recombinant protein production such as chloroplast expression can reduce the risk of spread of recombinant genes through pollens as chloroplast genes are maternally inherited in most plants. However, comprehensive and effective strategies will be necessary to prevent the environmental contamination of recombinant genes without severe negative effects on their yields, quality and processing.

## Figures and Tables

**Figure 1 ijms-23-13516-f001:**
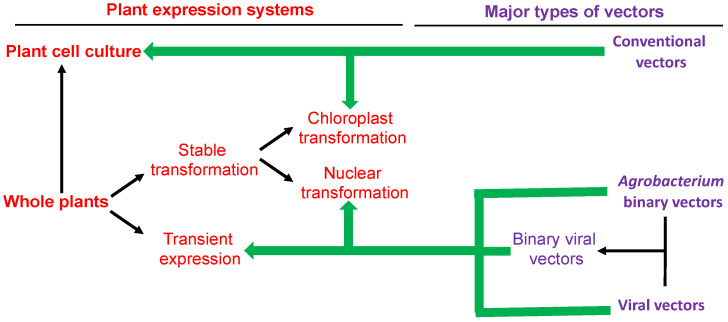
Major plant expression systems and vectors commonly used for production of recombinant proteins.

**Figure 2 ijms-23-13516-f002:**
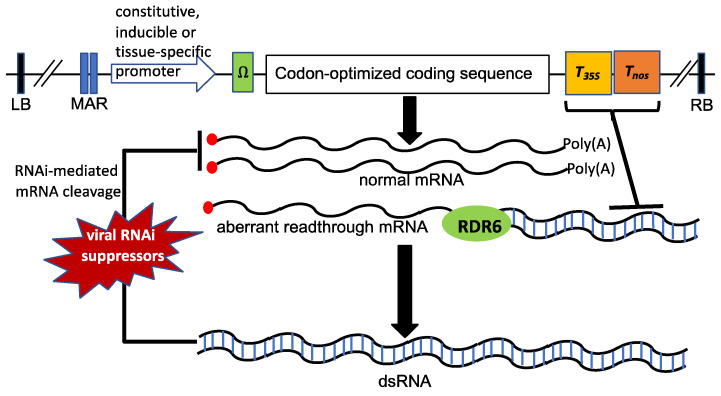
Major elements of the expression constructs for maximizing the production of recombinant proteins in plants through the enhancement of transcription and translation and the suppression of transcriptional and posttranscriptional gene silencing.

**Figure 3 ijms-23-13516-f003:**
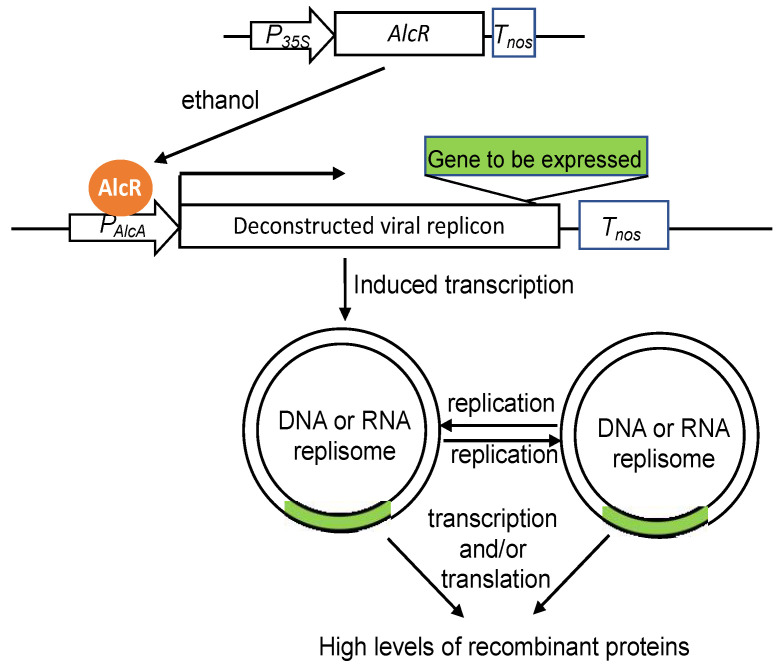
Inducible release of transgene-containing virus replicons for high level recombinant protein production in stable transgenic plants.

**Figure 4 ijms-23-13516-f004:**
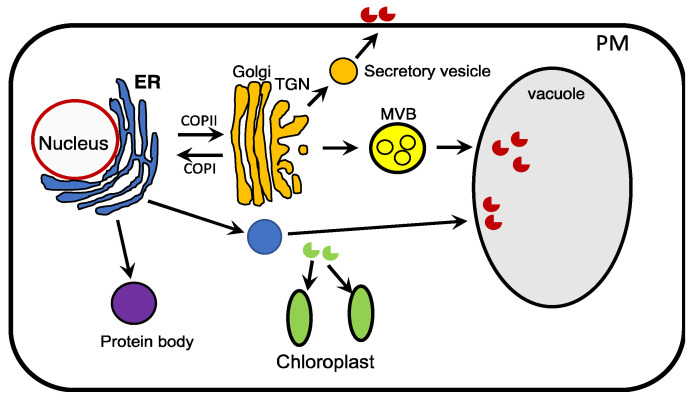
Targeting of nuclear-encoded recombinant proteins to different subcellular compartments for enhancing their production. ER-synthesized recombinant proteins can be targeted to the apoplast, vacuole, or protein bodies. Recombinant proteins synthesized in the cytosol can be targeted to chloroplasts with a N-terminal transit peptide.

## Data Availability

Not applicable.

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
