# Peer review of "Maximizing the Production of Recombinant Proteins in Plants: From Transcription to Protein Stability"

_ijms, 2022, doi:10.3390/ijms232113516_

Round 1

Reviewer 1 Report

This manuscript highlight the recent advances in the utilization of plants in the production of the recombinant proteins showing the effectiveness of plant cells as well as whole plants over other existing systems such as mammalian cells and bacterial expression systems. This work is relevant for a wide range of applications in food industry, biology and medicine. The authors eloquently elaborated on the topic by highlighting the different plant systems, vectors, that are currently used for the production of recombinant proteins, factors required to enhance and control the transcription and translation of the gene/transcript of interest, as well as some aspects of RNA sliencing mechnism and how to control it to improve the total production of the target recombinant protein. Although, and as one could expect from the title of this manusccript, the work lacks a section for a systematic comparison, for instance, between the production of some proteins between at least mammalian cells and plants, perhaps a Table showing a summary of some of the produced recombinant proteins in different expression systems and their characteristics would be sufficient to maximize the significance of the present work. Despite of this, the manuscript is extremely useful for the community and suitable for the publication in the IJMS after minor revision.

Author Response

We appreciate the reviewer's very positive comments and the suggestion of a list of produced recombinant proteins from different systems and their characteristics.  However, the number of the proteins produced in different systems is very larger and there is little information about their comparative characteristics since few proteins have been produced in all these different systems.  In the review, we have pointed out both advantages and disadvantages of plant-derived recombinant proteins.  In the revision, as requested by reviewer 2, we also pointed out that differences in protein postranslational modifications are also a major issue for some proteins.  Due to these issues, there are a relatively few therapeutic proteins that have been produced in plants and are currently on clinical trial.  We have included a recent review that contains a survey of these plant-derived therapeutic proteins and their status of clinical trial (lines 62-64).

Reviewer 2 Report

The review “Maximizing Production of Recombinant Proteins in Plants:  From Transcription to Protein Stability” focuses on various aspects of increasing the level of the heterologous protein production in plant. The authors compare various heterologous gene expression systems and note the advantages of plants as producers of therapeutic proteins and injectable and edible vaccines and proteins used in various industries. At the same time, they note a relatively small number of recombinant proteins derived from plants. One of the main reasons for this is the low level of synthesis and the high degree of degradation of target proteins.

It’s known that the production of the desired protein in a heterologous organism is a complex multistep process and involves several stages: cloning of the structural gene within the vector containing the promoter and terminator:  transformation and stable maintenance of heterologous DNA in the producing organism; gene transcription, and synthesis of the recombinant protein, and finally the purification and analysis of the resulting product. Among the main factors affecting the level of production of the heterologous protein are the copy number of the cloned gene, the transcription and translation efficiency, the accuracy of processing and folding of the synthesized protein, and its resistance to proteolysis.

The authors consider the advantage and disadvantage of main systems of heterologous genes expression in plants: plant cell culture and whole plants. They pay special attention to the expression vectors, mechanisms of regulation of transcription, translation and degradation of transcripts and target proteins.

Undoubtedly, the review will be useful for specialists in the plant genetics and biotechnology.  The review can be published with minor corrections.

A few remarks:

1.      It seems to me that in the review it should be noted that a certain disadvantage of plants as producers of mammalian glycoproteins is the difference in the structure of the carbohydrate component of such proteins in plants and mammals.

2.      Line 85: …”of pant systems and the DNA vectors…”.  Obviously should be”plant”.

3.      Lines 332, 342, 345: Why are the N. benthamiana underlined?

4.      Line 338: …”human alpha-1-antiryspin…”  Probably means alpha-1-antitrypsin (rAAT)?

5.      Line 361: …”AUT initiation codon…” Probably means AUG?

6.      Line 464: …”secretory proteins produced by filamentous fungi extraordinarily surface-active properties…”

Perhaps you should insert a preposition “with”: secretory proteins produced by filamentous fungi WITH extraordinarily surface-active properties…

Author Response

We appreciate the reviewer's very positive comments and very careful editing.  The reviewer pointed out that we should point out that differences in protein glycosylation between plants and animals might a issue for production of some therapeutic proteins in plants.  We have discussed the issue in the revised issue with a reference (lines 67-60).  However, the focus of the review is on the yield of recombinant protein production in plants and therefore, the issue of postranslational modifications is not extensively discussed unless they affect protein stability.

The reviewer has also pointed out a few typos and grammatical errors.  We appreciate the reviewer's editing and have corrected these errors in the revised manuscript.